# Honeybees Use Multiple Invariants to Control Their Altitude

**DOI:** 10.3390/insects14040313

**Published:** 2023-03-24

**Authors:** Aimie Berger Dauxère, Gilles Montagne, Julien R. Serres

**Affiliations:** The Institute of Movement Sciences, Aix Marseille University CNRS, ISM, CEDEX 09, 13284 Marseille, France

**Keywords:** optical invariant, ecological approach, ground-following task, optic flow

## Abstract

**Simple Summary:**

How do bees know they are changing altitude when they move close to the ground? It has been proved that humans use optical invariants, but their use remains unknown in insects. Anyway, the use of a single invariant, the optical speed rate of change, has been demonstrated in bees following the ground. Recently, it has been demonstrated that another invariant, the splay angle rate of change, could also be used by bees to adjust their altitude. This study aims to understand how bees use these invariants when they are available simultaneously. This issue has been addressed using an experimental tunnel providing discordant information to bees. We have shown that, when the two invariants were available, bees relied primarily on the optical speed rate of change to control their altitude. Conversely, when the optical speed rate of change was less easily accessible, the splay angle rate of change was used, unless bees perceive an imminent collision. Taken together, these results illustrate how the joint use of several invariants allows bees to navigate safely in unknown or cluttered environments.

**Abstract:**

How do bees perceive altitude changes so as to produce safe displacements within their environment? It has been proved that humans use invariants, but this concept remains little-known within the entomology community. The use of a single invariant, the optical speed rate of change, has been extensively demonstrated in bees in a ground-following task. Recently, it has been demonstrated that another invariant, the splay angle rate of change, could also be used by bees to adjust their altitude. This study aims to understand how bees use these invariants when they are available simultaneously. This issue has been addressed using an experimental setup providing discordant information to bees. We have shown that when the two invariants were available, bees performed ground-following tasks relying primarily on optical speed rate of change. Conversely, when optical speed rate of change was less easily accessible, splay angle rate of change was prioritized, unless the bees perceive danger. Taken together, these results illustrate how the joint use of several invariants allows bees to produce adaptive behaviors.

## 1. Introduction

In the middle of the last century, James Gibson put forward his seminal ecological approach of perception action [1]. According to his approach, all the information an agent needs to navigate in the environment is available in the perceptual flows and sufficient *per se*. According to the approach of J.J Gibson, the agent would have to detect the most relevant information to perform a given task. The visual flow is called Optic Flow (OF) and can be defined as a vector field of the angular velocities of objects, surfaces, and edges in a visual scene caused by the relative motion between the agent and the scene [2,3,4]. OF allows any moving agent to perceive their current relation with the environment. It provides information about the environment’s structure and about the displacement of the agent in relation to these structures [1,2]. Gibson’s framework has already proven its suitability in addressing issues related to the control of human displacement [5,6,7,8,9], as well as that of some birds [10,11,12], most flying insects [13,14,15,16,17,18,19,20,21,22,23], and robots [15,17].

OF contains low and high-order perceptual variables providing information of greater or lesser reliability about the state of the Agent-Environment System (AES) (e.g., in [24]). Low order variables are correlated to the state of the system; they allow an approximation of the AES without specifying it. Conversely, high order variables specify it, allowing a precise access to that state (see [25] for further information). The main characteristic of higher order variables is that whatever changes occur within the OF, certain styles of change are preserved. Gibson argued that the perceptual information necessary for guiding action is found in these higher-level variables, which are also known as invariants [2].

A number of invariants have already been shown to be used by agents in the control of goal-directed behavior. As an example, the first order temporal relation between an agent and an approaching object is specified by the relative rate of expansion of the optical contour of the object [26]. In this case, the agent does not need to know the size and speed of the object. They just need to detect the invariant (i.e., the relative rate of expansion) in order to precisely access the remaining time to contact. In another example, when an agent wants to intercept an object that is approaching along a parabolic trajectory, a convenient strategy is to move so as to cancel the optical acceleration of the object [27,28]. In this case, it is not necessary to know where and when the intercept will occur, just to detect the invariant (i.e., the optical acceleration) and modify the displacement accordingly.

As suggested by several authors [25], an alternative to the highly parsimonious invariant-based strategy could lie in the concomitant use of several invariants. This issue of multiple optical invariants use was addressed by Duchon and Warren [8] in an experiment carried out in humans. These researchers investigated the perceptual support used by participants to produce a centering behavior when moving through a virtual corridor. They analyzed the behavior of subjects when three invariants relevant for carrying out a centering task were either eliminated or distorted. The primary findings showed that these invariants were employed together in the control process; this prompted the authors to formalize a control law: (i) connecting the three invariants to a motion parameter and (ii) setting up a weight system governing the use of these invariants. When both are available, a combination process comes into play [29].

As many experiments support the idea that, when available, humans use several invariants in combination [6,9,30,31], we decided to investigate whether insects proceed similarly. Interestingly, previous research had provided results in agreement with the idea that flies would also use a combination of low and high-order variables for altitude control [23]. While it had already been proven that *Apis mellifera* uses an invariant, i.e., optical speed rate of change (OSRC), to control their altitude in a ground-following task [17,18,20], we have recently demonstrated that another invariant, i.e., splay angle rate of change (SARC), can also be used by bees to control their altitude in the same task [32]. Thus, the aim of the current study is to better characterize the simultaneous use of these two invariants in a ground-following task performed by bees.

## 2. Materials and Methods

For a flying agent performing a ground-following task, OSRC and/or SARC would unequivocally reveal a change in altitude. The setup we used (extensively detailed in [32]) was designed to uncouple these invariants and simultaneously record the flight paths of the bees.

### 2.1. Description of Two Altitude-Relevant Invariants

#### 2.1.1. Optical Speed Rate of Change (OSRC)

For a flying agent, objects scroll across its visual field (Figure 1). The optical speed of the ground (ω) is the ratio between the relative linear speed of the agent (*x*) and its altitude above the ground (*z*).

The optical speed of the ground (ω) can be calculated as follows:(1)$$\omega =\frac{\dot{x}}{z}$$

The closer the moving agent is to the ground, the faster the ground sweeps its visual field. For instance, on Figure 1, $\omega_{1}$ < $\omega_{2}$.

OSRC ($\dot{\omega }$) is the temporal derivative of the optical speed of the ground (ω). It can be calculated as follows:(2)$$\dot{\omega }=\frac{z\overset{..}{x}-\dot{x}\dot{z}}{z^{2}}$$

$\dot{x}$ is the agent’s forward speed and *z* is the agent’s altitude (Figure 1).

The optical configuration into our flight tunnel was designed to allow the uncoupling of the *x*-axis from the *z*-axis. It has been demonstrated that bees control their forward speed ($\dot{x}$) by referring to the smallest dimension of a tunnel (here, the width) [17,33]; using vertical stripes on lateral walls while manipulating the optical speed of the ground does not affect $\dot{x}$ [17]. As a result, the bee’s forward speed can be considered almost constant. Then, $\overset{..}{x}=0$. We can therefore simplify (Equation 2) as follows:(3)$$\frac{\dot{\omega }}{\omega }=-\frac{\dot{z}}{z}$$

As the width does not vary in our tunnel, a bee goes through the tunnel at a constant speed. As a consequence, any change in OSRC reveals a change in altitude.

ω is a low-order variable correlated to the altitude *z* of a flying agent. The link between ω and *z* is ambiguous as further speed–distance couples would result in the same ω. Meanwhile, $\dot{\omega }$, the temporal derivative of ω, is a high-order variable specifying the change in altitude $\dot{z}$ of an agent. These two variables are linked unambiguously as $\dot{\omega }>$ 0 always results from a loss of altitude $\dot{z}<$ 0. Inversely, an $\dot{\omega }<$ 0 always results from a gain of altitude $\dot{z}>$ 0 (See [32] for further details).

#### 2.1.2. Splay Angle Rate of Change (SARC)

When an agent flies over a flat surface, lines parallel to the direction of travel converge to a single vanishing point on the horizon. Splay angle *S* was defined by Flach et al. [34] as the angle subtended at the vanishing point by the direction of motion and the parallel lines (Figure 2). It can be calculated as follows:(4)$$S=arctan(\frac{y}{z})$$

The splay angle *S* is the angle subtended by imaginary lines in *y* to an agent of altitude *z*. As imaginary angles cannot be directly perceived, agents need to align them with objects or edges in their visual environment, such as hedges, paths, or vineyards, in order to gain access to them. Flach et al. [34] defined SARC as follows:(5)$$\dot{S}=\left(-\frac{\dot{z}}{z}\right)\times cosS\times sinS+\left(\frac{\dot{y}}{z}\right)\times cos^{2}S$$

SARC ($\dot{S}$) is the temporal derivative of splay angle (*S*), *y* is the lateral position, $\dot{y}$ is the agent’s lateral speed, *z* is the height above the ground, and $\dot{z}$ is the agent’s vertical speed (Figure 2).

As bees remain quasi centered in the narrow tunnel, we assume $\dot{y}=0$:(6)$$\frac{2\dot{S}}{sin\left(2\;s\right)}=-\frac{\dot{z}}{z}$$

A loss of altitude $\dot{z}<$ 0 gives rise to an increase in splay angle $\dot{S}>$ 0 (Figure 2), whereas an altitude gain $\dot{z}>$ 0 results in a decrease in splay angle $\dot{S}<$ 0 (Figure 2). Following Gibson’s reasoning, the splay angle *S* is a low-order variable correlated to altitude, and SARC $\dot{S}$ is a high-order variable specifying a change in altitude. Thus, to keep the same altitude, an agent has to negate any change in splay angle, i.e., to maintain $\dot{S}=$ 0 (See [32] for further details).

### 2.2. Materials

#### 2.2.1. Flight Tunnel

The flight tunnel had a rectangular shape (220 cm long, 71 cm high, and 25 cm wide). The entrance (a 5 cm diameter door) and the exit (5 × 5 cm) used were located 14 cm above the floor at their respective ends (Figure 3A). A 10 cm side box was situated behind the exit for the placement of a reward. The experimenter could manually manipulate both the entrance and exit from outside the tunnel.

#### 2.2.2. Pattern Providing the Optical Speed

The side walls, floor, and ceiling were textured with a printed pattern of red and white stripes (Figure 3A). On the ceiling and floor, the pattern was printed on one side of a plastic sheet (Figure 3(Bi)), while the other side was uniformly mat white (Figure 3(Biii)). These reversible sheets allowed us to provide greater or lesser difficulty in accessing optical speed (OS). With OS being generated by the scrolling of a point on a visual field, it is more accessible in contrast-rich visual environments. Thus, the striped plastic sheet offered, thanks to its strong contrast, an easy access to OS. The white sheet offered a difficult access to OS. It was, however, sufficient for the bees to perform a ground-following task [35,36]. In order to allow video recording, the striped pattern was reproduced with gelatin filter stripes on the left wall, which consisted of red and white stripes oriented perpendicularly to the insect’s flight path (See [32] for further details).

#### 2.2.3. Rods Providing Splay Angle

The flight tunnel enabled us to manipulate both splay angle and its rate of change (Figure 3B). A speed regulated DC motor operated a two-way metric screw located beneath the entrance and exit holes (Figure 3A). The screw held a green painted rod on each side that ran along the entire length of the tunnel at each floor–wall junction. The rotation of the screw either converged (Figure 3(Bii)) or diverged the rods. In the converging condition, the rods originally positioned at the floor–wall junction (Figure 3(Bi)) were set in motion in such a way as to give rise to an increase in splay angle. The diverging condition exposed the bees to the opposite stimulation, giving rise to a decrease in splay angle. Modifying the orientation of the rods between two successive passages was enough to manipulate the splay angle between flights.

#### 2.2.4. Video Recording and Flight Path Analysis

The bees’ trajectories were filmed at 100 frames per second with a 640 × 480 pixels resolution monochrome CMOS camera (Teledyne Dalsa Genie HM640, sensor size: 1/3″) fitted with a manually adjustable optical lens (Ref. H2Z0414C-MP, Computar, NY, USA) adjusted to a focal length of 4.9 mm. The camera was located at 1.52 m from the left wall (at 1.65 m from the tunnel midline), and its field of view (160 cm in width and 71 cm in height corresponding to 52° × 25°) covered the whole height of the tunnel, from abscissa *x* = 0 cm to abscissa *x* = 160 cm in all experiments. Image sequences were recorded with StreamPix 7 software (NorPix, Inc., Montreal, QC, Canada). They were then calibrated and corrected using a 3 × 3 cm checkerboard pattern of calibration (undistortImage, Matlab); then, they were cropped (size: 620 × 210 pixels) and analyzed using a custom-made Matlab program (The MathWorks, Inc., Netick, MA, USA), URL: https://github.com/rm1720/bees-applications, accessed on 6 February 2023). This program automatically determined the bees’ position (*x*, *z*), forming an ellipsis of 5 × 2 pixels in each frame, as a function of the time *t*. Next, by considering the ellipsis center in each frame, the bee’s trajectory in the vertical plane can be conveniently plotted.

The aperture of our optical lens is adjustable from F1.4 to F16. The depth of field is therefore sufficient to capture a sharp image of the bees’ lateral positions in our tunnel (see online Depth of Field Calculator, URL: https://www.dofmaster.com/dofjs.html, accessed on 6 February 2023). Our calibration procedure therefore generates a maximal error of 1 pixel, which vertically represents a maximum error of 3.4 mm. This latter error should be compared with the vertical parallax error, which can be estimated at 2 pixels by assuming the maximal oscillations of bees’ flight is ±3 cm around the recorded sagittal plane (see URL: https://www.youtube.com/watch?v=8WDcC-LmM9k, accessed on 6 February 2023), which, in total, represents a maximal error of 6.7 mm vertically.

Two-dimensional coordinates of bees were then sampled every 10 ms. Then, they were discretized through binning (binMed, Matlab). Each bin represents the median of 15 coordinates, meaning 150 ms.

### 2.3. Methods

In this research work, we explore the contribution of the two invariants in altitude control through a series of experiments in which the behavior of the bees has been analyzed in more or less impoverished visual environments, including conditions in which the two invariants provided contradictory information.

### 2.4. Familiarization Procedure

We worked with *Apis mellifera* circulating freely in the Parc National des Calanques (Figure 4). To allow them to become familiarized with the visual environment provided by the setup, a familiarization phase preceded each experiment. During this phase, bees freely travelled the tunnel in the control condition for 1 h.

During the test phase, the methodology used in this study involved recording the flights of bees across a series of successive trials that replicated the visual configuration of the control condition (catch trials). These catch trials were interspersed with a limited number of experimental trials in which a specific perceptual variable (either low oder or high order) was intentionally manipulated.

When we designed our experimental plan (See [32] for further information), we observed that the bees never took less than 3 min to come back to the setup (Figure 4). The manipulations took place in this time interval on different bees. Afterward, 10 min of control condition took place to allow the manipulated bees to get become accustomed to the visual control environment. Afterward, a period of 3 min of manipulations was repeated and so on. It is unlikely that naive bees that have never traversed the tunnel under control conditions will suddenly traverse it under test conditions in a straight line since they need training for this. The group of bees traversing the tunnel were obviously continuously sourcing on the reward placed at the exit of the tunnel until it was empty.

### 2.5. Predictions

When bees were able to use two invariants simultaneously to control altitude, several predictions could be made.

In the case of SARC being used, dynamic convergence of the rods should give rise to an increase in altitude, dynamic divergence should cause a decrease in altitude, while the control conditions should have no impact on bees’ altitude (Figure 5A). The converging condition was expected to result in an increase in altitude, since, in the absence of any violation of the laws of physics, this condition specifies a loss of altitude that must be compensated by an appropriate increase in altitude. The reverse is true in the diverging condition; the optically specified increase in altitude should be compensated by a decrease in altitude, whereas no altitude change was expected in the control condition.

The influence of SARC manipulations should be all the more pronounced because the floor is weakly textured. The presence of a textured floor should facilitate access to OSRC and lessen the influence of SARC (Figure 5A).

In the second experiment, the splay angle has been manipulated from one trial to the next by changing the angle of the rods. This manipulation influences the splay angle (low-order variable) while leaving the rate of change of the splay angle (high-order variable) unchanged. These manipulations should not give rise to systematic altitude changes (Figure 5B).

### 2.6. Statistical Analysis

The data did not follow a normal distribution (Shapiro–Wilk tests, *p* < 0.05) and did not respect homoscedasticity (Fligner–Killeen tests, *p* < 0.01). In experiment 1, Friedman tests (friedman.test R function) have been performed to test the influence of SARC manipulations on flight altitude. When main effects were found significant (*p* < 0.01), exact all-pairs comparisons tests of Friedman-type ranked data comparisons [37] (PMCMRplus package) were used to compare altitude binning distribution by pairs: bins before perturbation vs. bins after perturbation in the same condition (parallel condition before vs. after perturbation, converging condition before vs. after perturbation, etc.). In experiment 1, three bins before perturbation were compared to three bins after the perturbation. Pairwise comparisons between each bin in each of the control conditions (parallel or narrow) confirmed that bees’ altitudes were usually stable in the absence of perturbation. In experiment 2, Mann–Whitney tests have been performed to test the influence of splay angle manipulation on flight altitude (wilcox.test R function). Median altitudes of bees across the tunnel were compared to their control conditions. All statistical tests were performed using R software [38].

## 3. Results

### 3.1. Experiment 1: Joint Manipulation of Two High-Order Variables—OSRC and SARC

The aim of this experiment was to determine if bees use information from SARC and OSRC to control their altitude in a ground-following task. For this purpose, we manipulated SARC, while OSRC was either easily or hardly accessible. To manipulate SARC, rods were set in motion while bees were flying through the tunnel. To manipulate access to OSRC, two ground textures were used: striped texture provided an easy access to OSRC while white texture provided a difficult access to OSRC.

#### 3.1.1. Bees Undergoing a SARC Manipulation above White Ground Varied Their Altitude

In control conditions, no manipulation of SARC occurred. The results revealed that the bees passed through the tunnel without any change in altitude, either in parallel or in narrow control condition, above white ground (Figure 6A,B; for all comparisons *p* > 0.01).

In the converging condition (Figure 6A), the motorized rods were manually set in motion (blue dotted line) 2 s after the bees entered the tunnel. The results revealed that bees undergoing a positive SARC manipulation increased their altitude at 2.5 s and 2.75 s, i.e., respectively, 0.5 s and 0.75 s after the onset of the manipulation (Figure 6A and Table 1
*p* < 0.01 when compared to the altitude at 1.5 s and 1.75 s).

In the diverging condition (Figure 6B), the rods were also set in motion 2 s after the bees entered the tunnel, but this time in a divergent way. The results revealed that bees undergoing a negative SARC manipulation passed through the tunnel without any change in altitude above white ground (Figure 6B; for all comparisons *p* > 0.01).

#### 3.1.2. Bees Undergoing a SARC Manipulation above Striped Ground Did Not Vary Their Altitude

In control conditions (Figure 6C,D), no manipulation of SARC occurred. The results revealed that bees undergoing no SARC manipulation passed through the tunnel without any change in altitude, either in parallel or in narrow control conditions, above white ground (Figure 6C,D; for all comparisons *p* > 0.01).

In the dynamic converging and diverging condition (Figure 6C,D), the rods were set in motion (blue dotted line) 2 s after the bees entered the tunnel. The results revealed that bees undergoing SARC manipulations passed through the tunnel without any change in altitude either in converging or in diverging condition above striped ground (Figure 6C,D; for all comparisons *p* > 0.01).

### 3.2. Experiment 2: Manipulation of a Low-Order Variable—Splay Angle

The aim of this experiment was to determine if bees use SARC to control their altitude in a ground-following task or if they use splay angle, a low-order variable. To manipulate splay angle, the position of rods was manipulated between flights and did not move while bees were traversing the tunnel.

#### 3.2.1. Bees Undergoing a Splay Angle Manipulation above White Ground Did Not Vary Their Altitude

In control conditions (Figure 7A,B), no manipulation of splay angle occurred between flights. The results revealed that the bees undergoing no splay angle manipulation passed through the tunnel without producing a change in altitude, either in parallel or in narrow control conditions, above white ground (Figure 7A,B; for all comparisons *p* > 0.01).

In static converging and diverging conditions (Figure 7A,B), splay angle was manipulated between flights. The results revealed that the bees undergoing splay angle manipulations passed through the tunnel without any change in altitude, either in converging or in diverging conditions, above white ground (Figure 7A,B; for all comparisons *p* > 0.01).

#### 3.2.2. Bees Undergoing a Splay Angle Manipulation above Striped Ground Did Not Vary Their Altitude

In control conditions (Figure 7C,D), no manipulation of splay angle occurred between flights. The results revealed that the bees undergoing no splay angle manipulation passed through the tunnel without producing a change in altitude, either in parallel or in narrow control conditions, above striped ground (Figure 7C,D; for all comparisons *p* > 0.01).

In static converging and diverging conditions (Figure 7C,D), splay angle was manipulated between flights. The results revealed that the bees undergoing a splay angle manipulation passed through the tunnel without producing a change in altitude, either in converging or in diverging conditions, above striped ground (Figure 7C,D; for all comparisons *p* > 0.01).

## 4. Discussion

The aim of this study was to examine to what extent bees can use two invariants simultaneously to control their altitude while performing a ground-following task.

### 4.1. Bees Use SARC to Perform a Ground-Following Task

In a previous study [32], we investigated whether bees use SARC to perform ground-following task. The increase in altitude following the dynamic convergence of rods observed in this experiment constituted a first indication of the use of SARC by bees to control their altitude. In this study, manipulation of the orientation of the rods from trial to trial in the second experiment did not give rise to altitude changes. This result clearly indicates that it is less the angle of splay than its rate of change that is used by bees to control altitude. The bees’ use of SARC rather than the splay angle makes sense as perceptual strategies relying on high order variables are based on a precise access of the state of the AES and thus provide reliable strategies that lead to secure behaviors. It is worth mentioning that, in some cases, low-order variables can be preferred to high order variables [23,39]. For instance, *Drosophila* seems to prefer using a low-order variable to control its altitude in a flight tunnel [23]. This probably reflects the ability of insects to use the perceptual variables at their disposal. They are most likely to have several perceptual solutions to the problems they encounter.

### 4.2. Bees Combine SARC and OSRC to Perform a Ground-Following Task

Our second objective was to determine whether bees use a combination of high-order variables in a ground-following task and, if that is the case, to what extent and how this combination works. To that end, we put bees in a situation (which would be very unlikely in natural conditions) where OSRC and SARC provide contradictory information about the state of their relation to the environment. More precisely, in experiment 1, converging or diverging rods indicated a loss or a gain of altitude with reference to SARC, while OSRC did not suggest any change in altitude.

Interestingly, bees manage this informational conflict differently depending on the perceptual variable available. In the case of the floor having a strong contrast, providing bees easy access to OSRC, dynamic rod manipulations do not have an influence on bees’ altitude. Conversely, when the floor is weakly textured, making access to OS changes more difficult, the same rod manipulations give rise to an increase in altitude in the (dynamic) converging rod condition. In this case, the SARC is used to the detriment of OSRC.

From this point of view, the fact that the (dynamic) diverging condition with a white floor does not give rise to a decrease in altitude can appear counter intuitive. There may be a decrease in altitude; however, if so, it is too small to be detected by our statistical tests, which lack power given our small sample size and our use of non-parametric tests.

Assuming that (dynamic) diverging condition with a white floor does not give rise to a decrease in altitude due to experimental failure, we could hypothesize that, in the case of two invariants providing contradictory information to an agent about the state of its relation to its environment, priority is given to the invariant that results in the safest behavior [30,40]. In our case, using SARC would lead the bees to come dangerously close to the ground.

Taking all the results together, our study provides consistant results in favor of the joint use of several invariants by bees in their altitude control system. This strategy confers a certain robustness to the perceptual process. Bees can thus adapt their displacement within informationally degraded environments and produce the safest behaviors in the event of a conflict of inputs. It is also possible that individual behavioural variation leads some individuals to preferentially refer to one invariant rather than another; however, answering this question is beyond the scope of this study.

### 4.3. Is SARC alone Sufficient to Perform a Ground-Following Task?

In a seminal experiment, Heran and Lindauer observed that bees were unable to fly above a lake with a perfectly smooth surface [41], i.e, providing no access to OSRC. Wooden planks positioned perpendicular to the trajectory of the bees enabled them to do so. This result was said to demonstrate the primary role of OSRC in the control of altitude. With reference to the present study, it is worth mentioning that the addition of the wooden structure also allowed the bees to benefit from SARC. However, the contribution of Heran and Lindauer is remarkable and in the continuity of their findings, numerous studies have shown that bees are able to perform a ground-following task relying only on OSRC [17,18,20], which until now has seemed to be necessary for the achievement of this task. As an extension of our study, it would be interesting to explore to what extent SARC alone is sufficient to perform a ground-following task. Serres et al. [20] designed a specific configuration depriving the bees of any texture on the ground and ceiling of a tunnel and, as a consequence, from OSRC. Heran and Lindauer observed that bees were unable to fly through the tunnel. We hypothesized that, as we had investigated in the present study, the addition of rods materializing the splay angle would allow bees to detect SARC and, therefore, to perform the ground-following task successfully.

### 4.4. Learning Is a Matter of Educating Attention toward the Relevant Variable

The present study shows that, like humans, bees can employ several invariants simultaneously to perform a ground-following task. It would be interesting to explore the extent to which these invariants are combined (e.g., selection, vs., integration; relative weights attributed to invariants) and could change over repetitions of a task. Numerous studies conducted on humans have shown that learning leads to a change in the perceptual variable(s) employed, transitioning from lower order variable(s) to higher order ones [7,42]. From this information, the inability of bees to fly over untextured surfaces observed by Serres et al. [20] could perhaps be overcome thanks to repetitions in the presence of the splay angle. This would suppose that the bees are able to deprioritize (at least temporarily) the detection of the invariant preferably used (OSRC) in favor of another one (SARC). In any case, the similitude between perceptual-motor processes in humans and insects suggests that perceptual flexibility must also be considered in experimental designs involving insects.

## 5. Conclusions

The stated ambition of this research work is to bring to light the ability of bees to combine optical invariants to perform a given task. We show here that a priority system governs the use of optical speed rate of change (OSRC) and splay angle rate of change (SARC) in a ground-following task in bees. Our experimental observations reveal that bees use OSRC when it is available to perform a ground-following task and SARC when OSRC is difficult to access in the optic flow. Identifying the perceptual-motor principles underlying bees’ navigation and comparing it to that of humans would be useful in the design of flying aids and autopilots for drones in the near future. This conception of the perceptual processes opens the door to new prospections of research focusing on the underlying processes of combining perceptual variables in insects.

## Figures and Tables

**Figure 1 insects-14-00313-f001:**
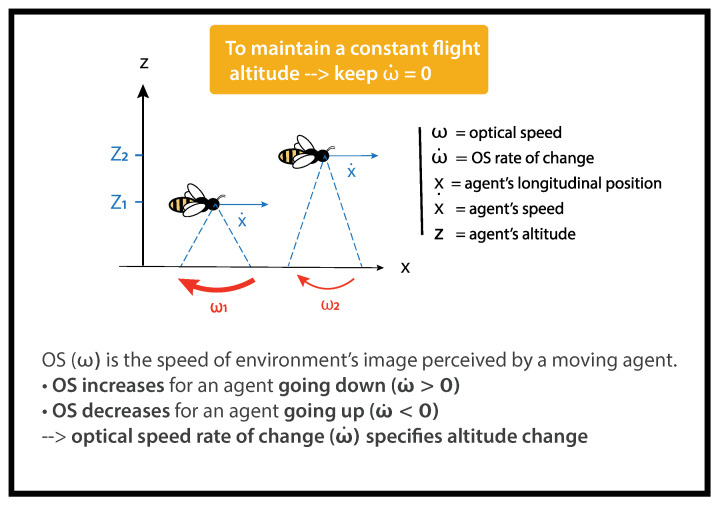
Optical speed (OS) and its rate of change (OSRC).

**Figure 2 insects-14-00313-f002:**
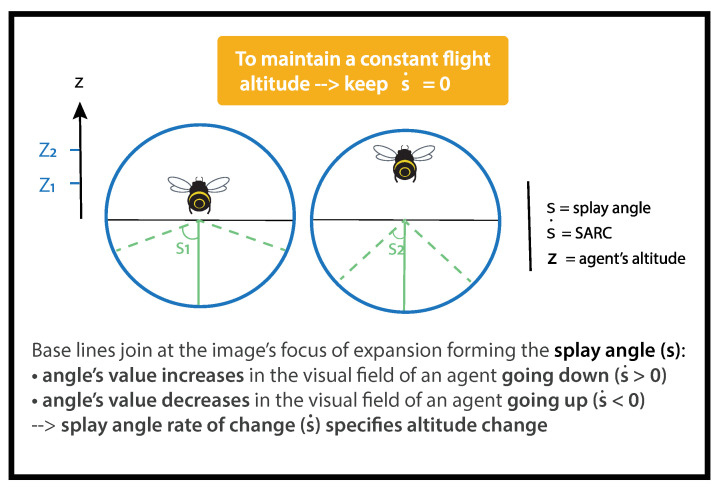
Splay angle and its rate of change (SARC).

**Figure 3 insects-14-00313-f003:**
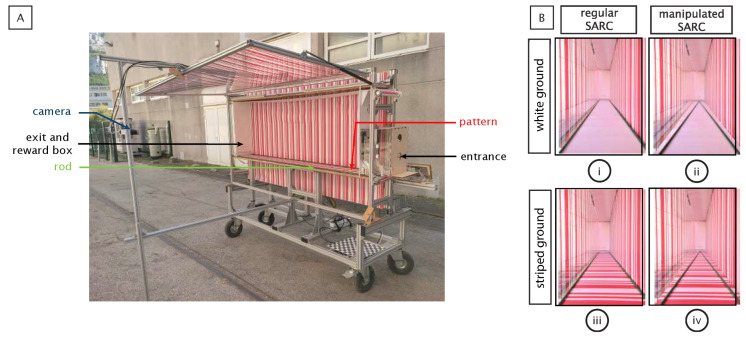
Internal and external views of the flight tunnel. (**A**) Flight tunnel designed to manipulate both SARC and OSRC. Textures on the ground could provide easy or reduced access to OSRC. Two motorized green rods allowed manipulation of either SARC or splay angle during or between the bees’ flights. Speed, acceleration, and amplitude of SARC could be tuned. (**B**) Optical invariants provided inside the tunnel. In (**Bi**,**Bii**) a uniform white sheet provided weak access to OSRC. In (**Biii**,**Biv**), a striped pattern provided easy access to OSRC. In (**Bi**), splay angle was not manipulated, whereas in (**Bii**), the rods were converging. SARC is positive from (**Bi**) to (**Bii**) and negative from (**Bii**) to (**Bi**). Adapted from [32].

**Figure 4 insects-14-00313-f004:**
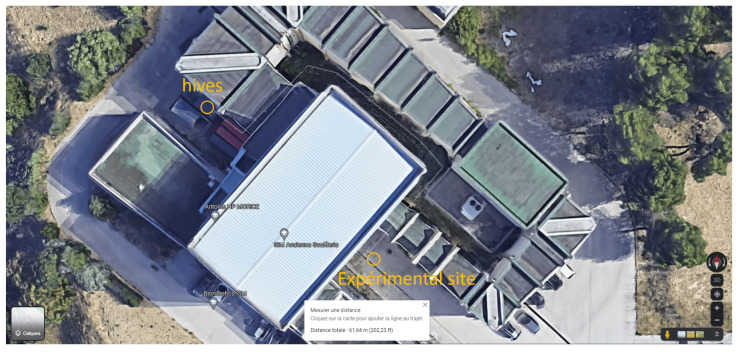
Geolocaltion of hives (43°14′02.9″ N 5°26′37.3″ E) and experimental site in the Parc National des Calanques, Marseilles, France. Both are separated by a 60 m path, and bees pass through it in at least 3 min.

**Figure 5 insects-14-00313-f005:**
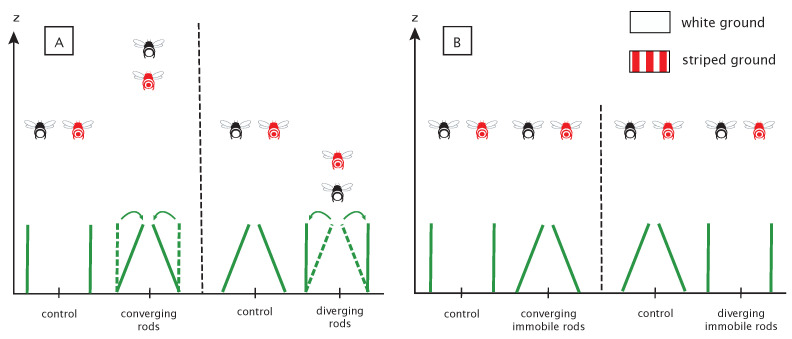
Predictive altitude of bees undergoing SARC or splay angle manipulations inside the flight tunnel. (**A**) Predictive altitude of bees during dynamic manipulations of SARC (experiment 1). (**B**) Predictive altitude of bees during static manipulations of splay angle (experiment 2). Red bees represent bees flying above striped ground and black and white ones represent bees flying above white ground. Immobile rods are the control conditions.

**Figure 6 insects-14-00313-f006:**
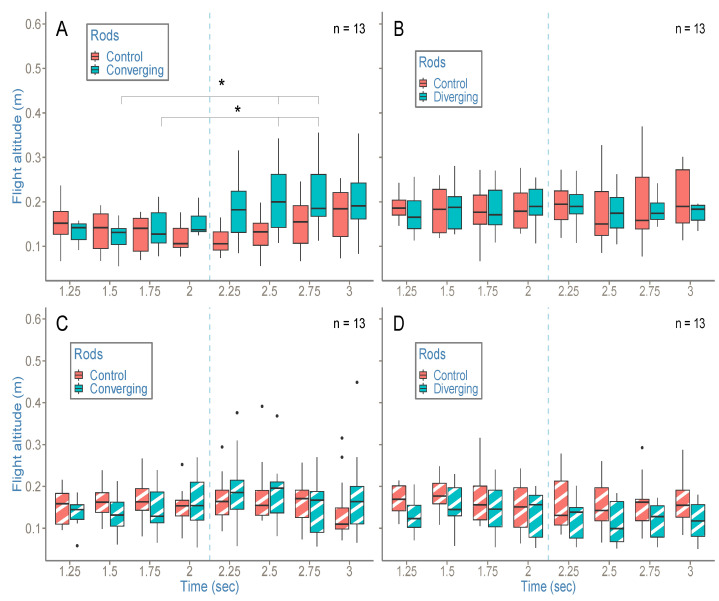
Altitude of bees subject to a splay angle manipulation. The four experimental conditions were (**A**) converging condition above white ground; (**B**) diverging condition above white ground; (**C**) converging condition above striped ground; (**D**) diverging condition above striped ground, compared to their respective control condition (parallel or narrow rods). The rods began to converge or diverge 2 s after bees entered the tunnel (blue dotted line), while they were flying through it. Above white ground, bees kept the same altitude in both the control conditions and the diverging condition (Figure 6A,B; for all comparisons *p* > 0.01). However, bees increased their altitude from 0.5 s to 0.75 s after the perturbation in the converging condition (Figure 6A; for all comparisons *p* < 0.01). Above striped ground, bees kept the same altitude in each condition and their respective control conditions (Figure 6C,D; for all comparisons *p* > 0.01). In all graphs, asterisks show significant differences between groups (*p* < 0.01) and outliers are represented by solid circles. Randomly selected flights were analyzed in each condition, respecting a ground-following criterion (see Section 2.2.4). Raw trajectories from the trials are visible in Appendix A. Data in (**A**,**B**) were already used in [32].

**Figure 7 insects-14-00313-f007:**
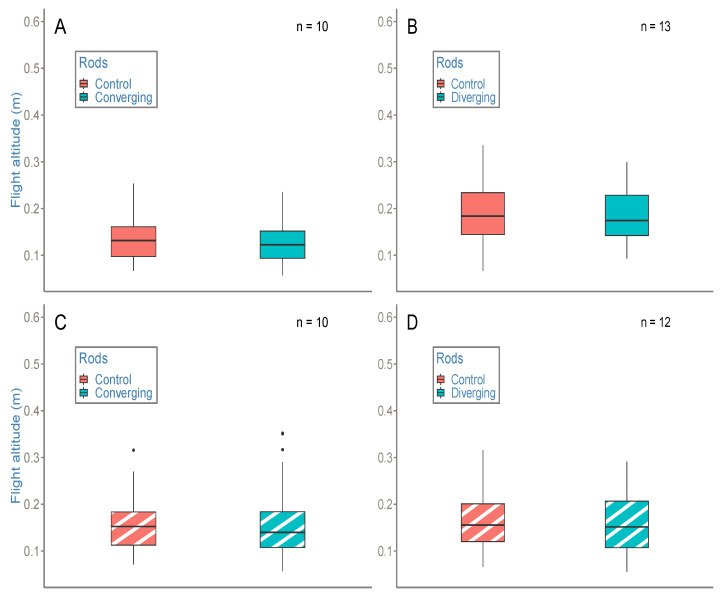
Altitude of bees subject to a change in splay angle value. The four experimental conditions were (**A**)—converging condition above white ground; (**B**)—diverging condition above white ground; (**C**)—converging condition above striped ground; (**D**)—diverging condition above striped ground, compared to their respective control conditions (parallel or narrow). The rods were set as converging or diverging before bees entered the tunnel. In each condition, bees maintained the same altitude for the whole of their flight through the tunnel (Figure 7; for all comparisons *p* > 0.01). Outliers are represented by solid circles. Randomly selected flights were analyzed in each condition, respecting a *ground-following criterion* (see Section 2.2.4). Raw trajectories from the trials are visible in Appendix A.

**Table 1 insects-14-00313-t001:** Results of the pairwise comparisons using the frdAllPairsExactTest R function. Bees undergoing a positive SARC manipulation increased their altitude at 2.5 s and 2.75 s, compared to 1.50 s and 1.75 s (*p* < 0.01, highlighted in bold).

	3 s	2.75 s	2.50 s	2.25 s	2 s	1.75 s	1.50 s
3s	-	-	-	-	-	-	-
2.75 s	1	-	-	-	-	-	-
2.50 s	1	1	-	-	-	-	-
2.25 s	1	1	1	-	-	-	-
2 s	1	0.10791	0.10791	1	-	-	-
1.75 s	0.14121	**0.00648**	**0.00648**	1	1	-	-
1.50 s	0.03385	**0.00096**	**0.00096**	1	1	1	-
1.25 s	1	0.38302	0.38302	1	1	1	1

## Data Availability

The data that support the findings of this study are openly available at https://github.com/abd34190/BergerDauxereetal2023.git, accessed on 28 February 2023.

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
