# Peer review of "Honeybees Use Multiple Invariants to Control Their Altitude"

_insects, 2023, doi:10.3390/insects14040313_

Round 1

Reviewer 1 Report

This study aims at disambiguating the role of two optical invariants in the altitude controls of honeybees. Those two invariants are the splay angle rate of change as well as the optical speed rate of change. The authors aim at answering this question by setting into conflict those two parameters in a flight tunnel. Both invariants were physically manipulated during the flight of an insect through a tunnel. The results showed that the optical speed was mainly used in the ground following task, except when this information was more difficult to access (white pattern). The author suggests that the splay angle rate of change is preferred in that specific case (with the white pattern), as long as it does not endangered the flight course.  

The introduction provides a good background. Nevertheless, a better definition of “invariants”, with clear examples could help a more naïve audience to understand the problematic of the paper.

l-155 In the video recording and flight path analysis: it is written that 13 trajectories were selected. It is not clearly explained why only 13 paths were selected. And how much this could affect the later results?

l-142 Using only one camera on the side as an estimate of altitude can lead to some error depending on the location of the bees in the flight tunnel. An estimate of this error would be a good add-on since the overall variations in height control is quite small (ranging around 10 cm on average of variations).

Figure 4-5 an inset showing all the trajectories overlayed with the average one could be informative since binning the data results in a lost of information. 

The individuals were not marked, there is therefore no possibility to know the effect of experience or inter-individuals’ differences in the following task. It would be of interest that the authors provide a reflection or justification on that point in the discussion. Indeed, there is quite a large variation when looking at the results that could be explain by individuals’ behavior. Maybe some individuals rely more on one invariant than the other. Making the current study only valid at a small populational level (because only 13 flights are used, that at best represents 13 individuals).

Finally, the authors should discuss that the tunnel itself seems to provide the bee with a “splay angle” due to its geometry (i.e. the pattern formed between the floor and the side walls). And reflect on how this can affect the relative low use of the splay angle in the present results.

The paper presents an interesting finding on how the rate of change in splay angle, that may be disregarded in some studies, impact flight control. In addition, the authors conclude that this invariant might be disregarded if leading to a dangerous information. However the conclusion misses a thorough discussion on the potential effect of the tunnel geometry to make their point stronger as well as the absence of individual marking.  

Reviewer 2 Report

This is an interesting study that uses a simple but clever method to independently manipulate two signals that bees could use to maintain flight altitude to evaluate whether and how they interact. The motivation and approach follow closely a previous publication, reference 29 in this paper, with the current paper expanding on the results provided there (which included the first direct demonstration that bees will change their flight altitude in response to a change in the convergence angle of lines parallel to their flight direction (splay angle rate of change, SARC)). The current paper compares this effect under conditions of stronger or weaker optical flow cues (stripes on the ground) given that optical flow (or specifically optical speed rate of change, OSRC) is already known to influence flight altitude in bees. It also reports the effect of static splay angles of greater or lesser degree. The conclusion is that bees will use SARC but only in one direction (to increase altitude but not decrease it) when OSRC is weak, but will not when OSRC is strong. The paper discusses in rather general terms how this indicates the joint use of several invariant cues to control behaviour. 

Main comments: 

  1. There is very substantial overlap with the previous paper: Figs 1 & 2, the majority of the materials and methods and the results in figure 4A. On one hand this is understandable as the same methods are used, the previous paper is clearly referenced, and it is helpful to have the details repeated here; on the other hand, there are places where it is not sufficiently clear. E.g. the only significant results (fig 4A) shown in this paper are the same results as shown the previous paper, and these are described on lines 264-265 as the current paper providing “a first indication of the use of SARC by bees to control altitude”. I am not sure what editorial policy might suggest here. 

  1. It was difficult to follow the intent, execution and analysis, of the second experiment. This is described in figure 5 as “bees subject to a change in splay angle value” where the value in any one trial was static but was varied (between two angles) between trials. First, is this the same bees experiencing different splay angles on successive trials (see more comment on this below)? This seems strongly implied by “the splay angle has been manipulated from one trial to the next, by changing the angle of the rods” (line 186) and later by “Bees undergoing a splay angle manipulation” (line 239), but if this could be for different bees, it does not really correspond to a ‘change’ for any individual.  Second, is there any difference between the control condition and the ‘static diverging rods’ condition? They appear the same in Fig 3B. Later in the caption it is mentioned that the ‘respective controlfor diverging and converging differs, i.e. is “parallel or narrow”. What does this mean, how do these controls differ from the static diverging and converging conditions, and why is it not illustrated in figure 3B? Third, if the intent is to test if bees use the (static) splay angle to control altitude, then surely the prediction would be that they should show a systematic difference in altitude between trials with different (static) splay angles, but perhaps only when OSRC is weak? That is, in fig 3B, the bee over white ground with static converging rods should have a higher altitude throughout the trial. But the analysis does not make this comparison, instead looking for a change in altitude within a trial. It seems obvious that no change should occur within the trial, as by design, the change in stimulus only occurs between trials. Maybe I have completely misunderstood the intent and method here but it certainly needs clarification. 

  1. Throughout the paper, it seems ambiguous whether the same bee experiences more than one condition, or experiences both control and (one or more) experimental condition. Are individual bees used as their own control, for example? This seems particularly relevant given the later suggestion (line 326-327) that bees might learn over time which variable to prioritize depending on their experience.  

  1. Many of the conclusions of the paper depend on interpreting a non-significant effect as the existence of no effect, with no discussion of the experimental power.  E.g the failure to see a decrease in altitude for diverging rods is explained as bees giving priority to the ‘safest’ behavior, rather than concluding that it could just be a smaller effect than for converging rods (possibly this follows from equation 6, and there is arguably a trend in 4B) which this experiment was unable to differentiate from noise. 

  1. I would like to have seen a more quantitative discussion of the ideas around cue integration, priority and weighting, particularly as it is a stated aim of the current study (lines 59-61) to characterize as precisely as possible the combination process underlying the simultaneous use of these two OF invariants. The somewhat vague conclusion (lines 334-335) that a priority and weight system governs the use of optical speed rate of change (OSRC) and splay angle rate of change (SARC) does not really meet this aim. E.g. how would a “priority and weight” system be distinguished from a priority system alone vs. a weight system alone? 

Minor comments: 

Line 4-5: it seems unfair and unnecessary to say that the concept of invariants is “little known in the entomology community” especially as it seems directly contradicted by the following sentence highlighting an invariant that this community has studied extensively. 

Line 93 Spay -> Splay 

Line 154-157 The paper describes taking a ‘sample’ of the available trajectories to analyse for a particular condition, why not use all the data? 

Line 193 “Individual identity was included as a random factor” - is every trajectory from a different bee or are their mixed independent and repeated measures in this design, and would including identity as a factor sufficiently address this in the linear mixed model? 

Line 194-200 I am not convinced that multiple pairwise comparisons are the appropriate way to analyse this data, which is essentially a time series. Perhaps the approach could be explained in more detail? 

Reviewer 3 Report

MS Number: insects-2135143

MS Title: " Combination of optical invariants in honey bees’ altitude control"

Review Report: 

This study addresses an important topic related to honey bee flight navigation and perception. Authors attempted to untangle honey bee reliance on either SARC and/or OSRC in flight altitude control. This was done by manipulation of the splay angle and ground rod motion as described in the M&M. The manuscript is well written, and the data is relevant. That said, there are some minor modifications that I strongly recommend, which will provide more clarity to this complex topic. 

Minor issues: 

1-    Abstract: please revise the abstract to make it more specific and descriptive of your findings.  Currently, it is too vague with long historic background that is not necessary in scientific abstract, e.i: mention of J.J Gibson,  L3: “Anyway”: delete. L8: “ the ambition” : delete, use of “insect”: no need to generalize, this work was conducted on honey bees…etc ,Reshape and sharpen all that and go straight to the point.  

2-    Fig. 5: the blue dotted lines are missing. 

3-    Nothing is significant in this figure too, there is no asterisks, delete the last sentence of the Figure’s caption. 

4-    L269: “..not the splay angle by makes sense…” correct please, there seems to be a mistake here, rephrase. 

Data to add/explain: 

1-          It is not clear why the authors decided on a statistical cut-off of P<0.01 and not a regular 5%? This should be stated. 

2-    It is not mentioned if the data (box plots) followed a normal distribution of not? 

3-    3- How do you explain presence of many outliers? 

4-    Why authors decided on n=10, 12, 13 replicates only?? Once your tunnel was up and running, it would not have been very difficult to run much higher number of replicates, which would have helpt stabilizing the dataset and curied the outliers. 

5-    Honey bees: L 164: “circulating freely”: these were foragers, which means they significantly differ in age and behavior, older ones would usually be more experienced than newly foraging bees. This is a significantly random factor that was only left to the Lmer model to deal with as a “random factor” apparently, but do you think this is satisfactory?

6-    The tunnel: elaborate on the light/luminosity accessible in the tunnel. Was the exit point open to the outside?  How did you account for the potential bees’ attraction to the light source at the end of the tunnel? This might have been a major driver of the bee’s navigation and altitude control? These points should be tackled/resolved in the text.        
